# A high-resolution gridded dataset of daily temperature and precipitation records (1980 – 2018) for Trentino – South Tyrol (north-eastern Italian Alps)

Alice Crespi[1], Michael Matiu[1], Giacomo Bertoldi[2], Marcello Petitta[1,3], Marc Zebisch[1]

[1]Institute for Earth Observation, Eurac Research, Bolzano, 39100, Italy
[2]Institute for Alpine Environment, Eurac Research, Bolzano, 39100, Italy
[3]SSPT-MET-CLIM, ENEA, Rome, 00196, Italy

*Correspondence to*: Alice Crespi (alice.crespi@eurac.edu)

**Abstract.** A high-resolution gridded dataset of daily mean temperature and precipitation series spanning the period 1980 – 2018 was built for Trentino – South Tyrol, a mountainous region in north-eastern Italy, starting from an archive of observation series from more than 200 meteorological stations, covering the regional domain and surrounding countries. The original station data underwent a processing chain including quality and consistency checks, homogeneity tests, with the homogenization of the most relevant breaks in the series, and a filling procedure of daily gaps aiming at maximizing the data availability. Using the processed database, an anomaly-based interpolation scheme was applied to project the daily station observations of mean temperature and precipitation onto a regular grid of 250 m x 250 m resolution. The accuracy of the resulting dataset was evaluated by leave-one-out station cross-validation. Averaged over all sites, interpolated daily temperature and precipitation show no bias, with a mean absolute error (MAE) of about 1.5 °C and 1.1 mm and a mean correlation of 0.97 and 0.91, respectively. The obtained daily fields were used to discuss the spatial representation of selected past events and the distribution of the main climatological features over the region, which shows the role of the mountainous terrain in defining the temperature and precipitation gradients. In addition, the suitability of the dataset to be combined with other high-resolution products was evaluated through a comparison of the gridded observations with snow-cover maps from remote sensing observations. The presented dataset provides an accurate insight on the spatio-temporal distribution of temperature and precipitation over the mountainous terrain of Trentino – South Tyrol and a valuable support for local and regional applications of climate variability and change. The dataset is publicly available at https://doi.org/10.1594/PANGAEA.924502, Crespi et al. (2020).

## 1 Introduction

High-resolution gridded datasets of in situ climate observations are of increasing relevance not only for the studies on climate and its variability but also for many applications, such as natural resource management, adaptation planning, modelling and risk assessment in a wide range of fields including hydrology, agriculture and energy (Haylock et al., 2008; Hofstra et al.,

2008). For instance, spatialized in situ observations can be used in runoff, crop growth and glacier mass balance modelling (Engelhardt et al., 2014; Ledesma and Futter, 2017), to validate and bias-correct climate simulations (Kotlarski et al., 2019; Navarro-Racines et al., 2020), to calibrate and integrate remote sensing products (Schlögel et al., 2020) and to develop advanced monitoring systems supporting decision making (Aadhar and Mishra, 2017). This is particularly meaningful for local studies over topographically complex areas experiencing a high climatic heterogeneity, such as the Alps, where the Italian

region of Trentino – South Tyrol is located. Mountain areas are essential for freshwater and hydropower production, and they are particularly prone to natural hazards such as floods, landslides and avalanches which threat human life and infrastructures and require monitoring and prevention (Immerzeel et al., 2020).

    Several gridded products of temperature and precipitation series derived from in-situ observations are currently available globally, at European and at national scales at different temporal and spatial resolutions, for example CRU at ~ 50 km

horizontal spacing for global monthly data (Harris et al., 2020), E-OBS for Europe at ~ 10 km resolution (Haylock et al. 2008), GAR-HRT (Chimani et al., 2013), APGD and LAPrec (Isotta et al., 2014; Isotta et al., 2019) for Alps at around 5 km grid spacing. The datasets are derived by applying interpolation methods to the meteorological station records unevenly located over the territory to obtain estimates for each point of a regular grid. The accuracy of the results depends on the available station coverage, which could be particularly sparse for some mountain areas. However, the spatial resolution of the large-

scale gridded products (from few to tens of km) does not allow to properly capture the finer climate gradients of mountainous regions such as Trentino – South Tyrol and to respond to the needs of local applications. Further gridded data were computed at transregional, sub-regional and catchment levels at different timescales and at finer resolution, generally 1 km horizontal spacing (e.g. Brunetti et al., 2012; Laiti et al., 2018; Mallucci et al., 2019), however most of them cover only partially the study region and/or are not up to date, which limits their applicability for operational purposes. In addition, several global and quasi-

global precipitation products based on satellite only or on satellite-gauge combination have been developed in the recent decades at different temporal and spatial resolutions, such as GPCP (Adler et al., 2018), TRMM (Huffman et al., 2007) and CHIRPS (Funk et al., 2015), whose performances have been recently evaluated with respect to ground station data over parts of the study region (Mei et al., 2014; Duan et al., 2016). However, most products are available at 0.25° grid spacing and further enhancements in spatial resolution and agreement with daily observations are still needed to improve their application in

smaller basins (Duan et al., 2016).

    The choice of the horizontal resolution of the gridded products is handled differently by the different communities. From a climatological point of view, developing kilometre scale or even sub-kilometre scale products based on interpolated sparse observations does not provide more information than deriving coarser products (Haylock et al., 2008). The effective resolution is mostly defined by the underlying station distribution and it can be different from the target grid spacing (Grasso, 2000;

Lussana et al., 2019). However, from a user perspective, higher resolution data might be more desirable, because they are closer to the actual problems for practical applications (Beven et al., 2015), requiring a precise definition of local gradients which in mountainous terrains can occur over distances less than 1 km. Even if they contain similar information as their coarser resolution counterparts, a sensible interpolation approach of climate variables to finer spatial scales is beneficial, especially in

highly complex mountain terrains such as Trentino – South Tyrol, to properly account for the orographic gradients in a wide range of applications, e.g. modelling of snow, hydrological processes, vegetation or heat-related health impacts. Moreover, high-resolution data can be used more conveniently in hydrological models or for processing satellite observations without the need of a further downscaling.

Different interpolation techniques have been developed so far to derive gridded climate products and the choice largely depends on the domain features, data availability and desired spatial details. The proposed methods include inverse distance weighting (IDW, Camera et al., 2014), splines (Stewart and Nitschke, 2017), geostatistical schemes such as kriging and its variants (Hengl, 2009; Sekulić et al., 2020), optimal interpolation (Lussana et al., 2019) and regression-based approaches (Daly et al., 2007; Brunsdon et al., 2001). In topographically complex domains the interpolation schemes modelling the relationship between the terrain features and the climate gradients are preferable (Daly et al., 2002). However, the spatialization methods are required to deal with the heterogeneity of observation availability, which typically decreases e.g. in high-elevation mountain regions, where only over recent decades new automatic meteorological stations have been settled and provide information for previously uncovered areas.

Geostatistical and regression-based methods computing local climate gradients require dense data coverages and could provide unreliable spatial patterns where data density is low and strong spatial variability occurs, especially for precipitation fields and at daily timescale (Hofstra et al., 2010; Ly et al., 2011; Crespi et al., 2018). More straightforward approaches, such as IDW, could provide more stable results even though the final spatial variability could result highly smoothed and affected by over/underestimations if data are unevenly distributed (Di Piazza et al., 2011). In order to partially overcome these issues, anomaly-based methods were proposed in which the final gridded daily distribution for a certain variable is obtained by superimposing the long-term climatological values of reference, generally 30 year means, and the spatial distribution of the local daily deviations from them (e.g. New et al., 2001). Especially for daily precipitation, the interpolation with a reference field and anomalies was proved to be less prone to errors than the direct interpolation of absolute values, such as systematic underestimations in high-mountain regions due to the prevalence of stations located in the low valleys (Isotta et al., 2014; Crespi et al., 2021). This concept was applied in a relevant number of studies (see e.g. Haylock et al., 2008; Brunetti et al., 2012; Chimani et al., 2013; Hiebl and Frei, 2018; Longman et al., 2019).

In this work, we present the gridded dataset of daily mean temperature and precipitation for Trentino – South Tyrol region covering a 39 year period (1980 – 2018) at very high spatial resolution (250 m). The gridded dataset is computed from a collection of more than 200 daily station records retrieved from the regional meteorological network and checked for quality and homogeneity. The daily interpolation is based on the anomaly concept by superimposing the 1981 – 2010 daily climatologies at 250 m resolution computed by a weighted linear regression with topographic features and the 250 m resolution fields of daily anomalies obtained by a weighted-averaging approach. To the authors' knowledge, this is the first time such interpolation scheme is applied to derive daily fields of these two key climatic variables at such a fine grid spacing. The resulting gridded climate product includes the small-scale terrain features of the region and can be easily integrated and combined with models and other high-resolution data, e.g. remote sensing observations, without applying any further

downscaling to get a target sub-kilometre scale. In addition, the resulting product was designed to be regularly updated by the recent station records in order to respond to more operational purposes.

Section 2 describes the collection and processing of the meteorological station data and provides a detailed explanation of the applied interpolation technique. In Sect. 3 the accuracy of the gridded dataset is discussed on the basis of the results of cross-validation analyses, and the regional climate features and selected examples derived from the computed fields are presented. In addition, a preliminary comparison of the 250 m gridded temperature and precipitation series with the 2001 – 2018 winter snow cover data from remote sensing images over the region is reported in order to provide an example of the potential

applications of the dataset in combination with other high-resolution products. Section 4 provides the information about the dataset availability and access. Finally, the summary of the main outcomes and outlooks of the work are reported in Sect. 5.

## 2 Data and Methods

### 2.1 The study area

Trentino – South Tyrol is a region located in north-eastern Italy covering an area of around 13,000 km$^2$ (Fig. 1). Its territory is

entirely mountainous, including a large portion of Dolomites and southern Alps and it is located at the intersection region for various types of air masses: humid influences from the Atlantic northwest, dry air masses from the continental east, typically experiencing cold winters and warm summers, and warm contributions from the Mediterranean area bringing humid winters and dry summers (Adler et al., 2015). The geographical location and the complex topography of the region determine a strong climate variability and contribute to define small-scale effects in the spatial distribution of temperature and precipitation (Price,

2009). The territory is characterized by strong altitude gradients with very narrow valleys surrounded by steep slopes. Elevation range extends from 65 m a.s.l. in the areas close to Lake Garda in the south to 3905 m a.s.l. of the Ortler peak in the Stelvio National Park (north-west). The mean elevation of the region is around 1600 m with only 4% of flat areas, i.e. with slope steepness below 5%. The main valley is the one of Adige River, the second longest Italian river, flowing from Reschen Pass in the north-western corner of the region, crossing Venosta Valley west to east and then the entire region north to south towards

the Adriatic Sea.

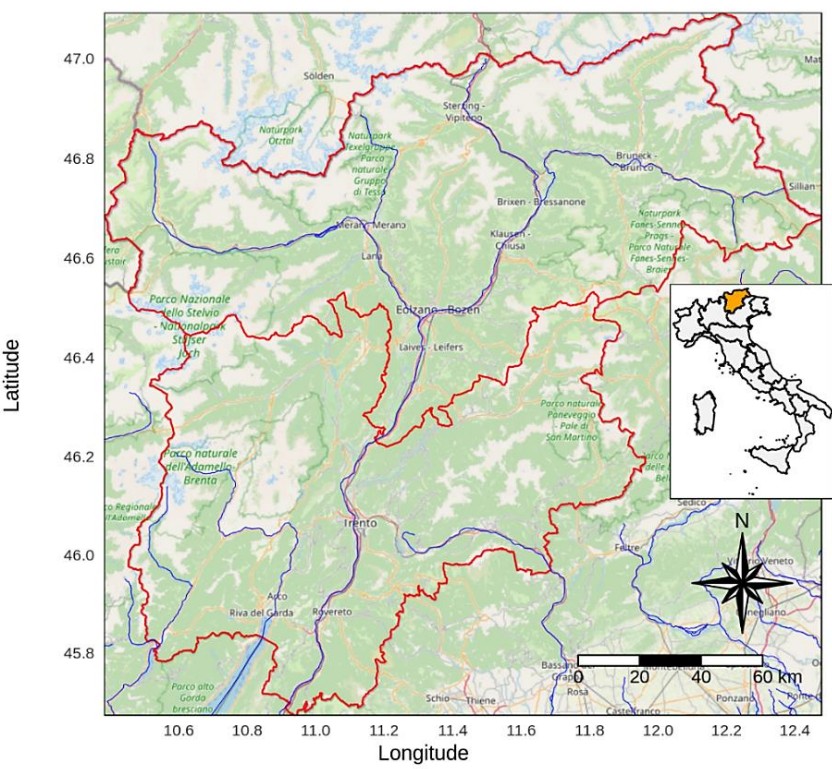

**Figure 1: The study region of Trentino – South Tyrol composed by the Autonomous Provinces of Trento and Bolzano (red bordered) and the region location in Italy (orange area in the inset plot). [© OpenStreetMap contributors 2020. Distributed under a Creative Commons BY-SA License]**

## 2.2 The observation database

The database used for the present study was set up by retrieving the observation series of daily maximum and minimum temperature ($T_{max}$, $T_{min}$) and daily total precipitation (P) from the stations of the meteorological networks of Trentino – South Tyrol region. More precisely, the data for the Autonomous Province of Trento were received from Meteotrentino (https://www.meteotrentino.it/), while the meteorological series for the Autonomous Province of Bolzano were provided by the Hydrographic Office. In total 311 series of daily $T_{max}$ and $T_{min}$ and 243 series of daily P were collected (Fig. 2). The series were integrated with the records from few close sites located in surrounding regions, especially in the north where the terrain is mainly mountainous, and the less dense regional coverage could not be enough to provide a suitable representation of climate gradients in the high-Alpine environment over the borders. At this aim, some daily series from Switzerland, Austria and Veneto were also included. Swiss sites were retrieved from MeteoSwiss (IDAWEB, https://gate.meteoswiss.ch/idaweb/), Veneto data were provided by the Regional Agency of Environmental Protection (ARPA Veneto), while Austrian series were provided by the Zentralanstalt für Meteorologie und Geodynamik (ZAMG). In addition, data for some further Austrian locations were

collected from the HISTALP database of daily homogenized records (Auer et al., 2007; http://www.zamg.ac.at/histalp/dataset/station/csvHOMSTART.php).

The control of data quality and homogeneity represents a crucial preliminary step to improve the general accuracy of the collected observation series. The weather station records can be affected by a number of both random and systematic errors, including erroneous transcriptions of data, station malfunctions, sensor drifts and inhomogeneities due to non-climatic factors, such as station relocation, changes in surroundings or in employed sensors. From around the 1990s, the manual measurements were replaced by automatic stations in most meteorological services. In this study we did not distinguish between manual and

automatic records and their consistency was assured by quality-check and homogenization procedures. In addition, the observation time and meteorological day definition slightly vary among the networks and could change within the years, especially in the shift from manual to automatic systems. For instance, daily precipitation records from the regional network are generally referred to the cumulated precipitation from 08:00 UTC of previous day to the 08:00 UTC of current day and also the recent automatic records are defined by following this definition. However, this information is not always reported,

and other daily reading times could have been adopted in some cases such as 07:00 UTC and 00:00 UTC. Clear shifts due to changes in time coding were checked and corrected, but no specific correction of the observation time was applied and induced inhomogeneities were assessed through the quality-check procedures.

The collected database spanned the period from 1950 to present and the whole interval was used for the quality and homogeneity checks in order to increase the statistics and the robustness of the analyses. The series in close proximity

(horizontal distance < 3 km and vertical distance < 100 m) but covering different time periods were merged in order to improve the length and continuity of available records. The merging concerned mostly some stations for Trentino province, which were split due to slight relocations and/or after the transition from mechanical to automatic sensors.

The quality-check analyses were performed on the daily series of $T_{max}$, $T_{min}$ and P in order to detect outliers and to assess the spatial and temporal consistency of data. Implausible values, such as negative precipitation and out-of-range records, were

scanned by setting fixed thresholds. In particular, P exceeding 500 mm, $T_{min}$ < -40 °C, $T_{min}$ > +40 °C, $T_{max}$ < -30 °C, $T_{max}$ > +50 °C and diurnal temperature range > 35 °C were considered and removed if not supported by metadata or by surrounding station records. In addition, cases of daily $T_{min}$ exceeding $T_{max}$ were invalidated and periods of continuous null daily P over more than one month were detected and removed if a simultaneous dry period was not reported for nearby sites. Temporal consistency was also scanned in both $T_{max}$ and $T_{min}$ series by searching for differences in consecutive days greater than 20 °C

(Durre et al., 2010).

In order to further assess the overall accuracy of the series, monthly records were computed and each one was simulated over the whole spanned period by means of the surrounding station data (Crespi et al., 2018, Matiu et al., 2021). The agreement between monthly observed and simulated values was measured in terms of bias, MAE and squared correlation ($R^2$) and allowed to assess the spatial consistency of the series and to further detect stations affected by frequent malfunctions and periods of

suspicious records. In particular, 11 temperature series and 13 precipitation series with low-quality data or duplicates of other series were identified and discarded from the following analyses. The final mean reconstruction bias over all $T_{max}$ and $T_{min}$

series was almost null, mean MAE was 0.7 °C and 0.6 °C, respectively, with a mean $R^2$ of 0.98 for both variables. For P series, the average relative bias was -1%, the relative MAE 17%, and the mean $R^2$ was 0.89.

After the quality check, all series with more than 30 years of valid records underwent homogeneity controls and the ones
showing relevant breaks were homogenized. To this aim, the Craddock test (Craddock, 1979) was applied by following an approach similar to that adopted in previous studies (see e.g. Brunetti et al., 2006; Brugnara et al., 2012). The homogeneity test was applied to the monthly records by using as reference the 5 nearby stations with the highest number of data in common. In some cases, the comparison with the available homogenized records from HISTALP and MeteoSwiss supported the identification of possible breaks. The T and P records showing relevant breaks were homogenized by deriving the correcting
factors at monthly scale and applying then the adjustments to the daily values. More specifically, once an inhomogeneous period was identified in the test series, the adjustment for each month was estimated independently as the arithmetic average of the corrections derived from each reference series according to a common homogeneous period. In the case of precipitation, if no annual cycle was evident a single yearly adjustment was computed as the average of the monthly factors, otherwise the monthly factors were used. The most recent years were always left unchanged in order to ease the update of station records.
For precipitation series, the corrections derived from the monthly values were directly applied as multiplicative factors to the daily records in the corresponding inhomogeneous period. In the case of temperature series, in order to take into account the annual seasonality and extract a correction for each day of the year, the 12 monthly factors were interpolated at daily resolution by means of a second-order trigonometric interpolation. The resulting 365 (366 for leap years) daily adjustments were then applied to the inhomogeneous daily records as additive corrections.
If $T_{max}$ and $T_{min}$ series were corrected, their internal consistency was checked to be preserved. In total 11 $T_{max}$ series, 13 $T_{min}$ series and 48 P series were homogenized. Short periods with strong breaks for which no robust correction factors could be computed were invalidated. A total of 20 and 22 breaks in $T_{max}$ and $T_{min}$ were identified with almost half of them concerning the years prior to 1980s, which is outside the period of the computed spatial product. The average length of the inhomogeneous periods was about 7 years with an average annual adjustment of +1.2 °C and -1.1 °C for $T_{max}$ and $T_{min}$, respectively. As regards
the P series, 84 breakpoints were identified in total and the mean annual multiplicative factor applied was about 1.1. Similar to T series, the average extent of the corrected periods was about 11 years and they mainly occurred before the 1980s. Due to the lack of specific metadata supporting the break interpretation in the series, we adopted a precautionary approach and no correction was performed if the breakpoint could not be identified clearly.

As last step, in order to increase the data coverage and to maximize the temporal extent of the records, a gap-filling procedure
was applied to reconstruct the missing daily data. The filling was performed by considering the data of the surrounding station with the highest correlation with the data of series under evaluation and by rescaling its daily value for the ratio (in case of precipitation) and the difference (in case of temperature) of the daily averages of the two series over a common subset of data, which is defined by a window centred on the daily gap and extending over both years and days. The reconstruction was performed only if the test series contained at least 70% of valid daily data in the selected window and a total of around 8000
daily entries was reconstructed over the period 1980 – 2018.

All processed series with less than 10 years of records were discarded from the database used for the interpolation since they could not provide robust long-term references from which the anomalies were computed. Daily mean temperature series were finally derived as the average of $T_{max}$ and $T_{min}$ values.

The resulting database contained 236 daily mean temperature and 219 daily precipitation series, 205 and 188 out of them located within the region, respectively (Fig. 2). The station distribution varies strongly with elevation. The best data coverage is between 500 and 1500 m for both temperature and precipitation databases, while the number of available stations gradually decreases for higher altitudes and no precipitation sites are located above 3000 m. The portion of the domain above 1500 m is, in relative terms, underrepresented by the available observations thus a larger uncertainty needs to be assigned to the final gridded estimates for such areas (Fig.3). The data coverage over higher altitudes can be improved in the future thanks to the integration of the most recent automatic weather stations when their temporal extent becomes suitable for climatological assessments.

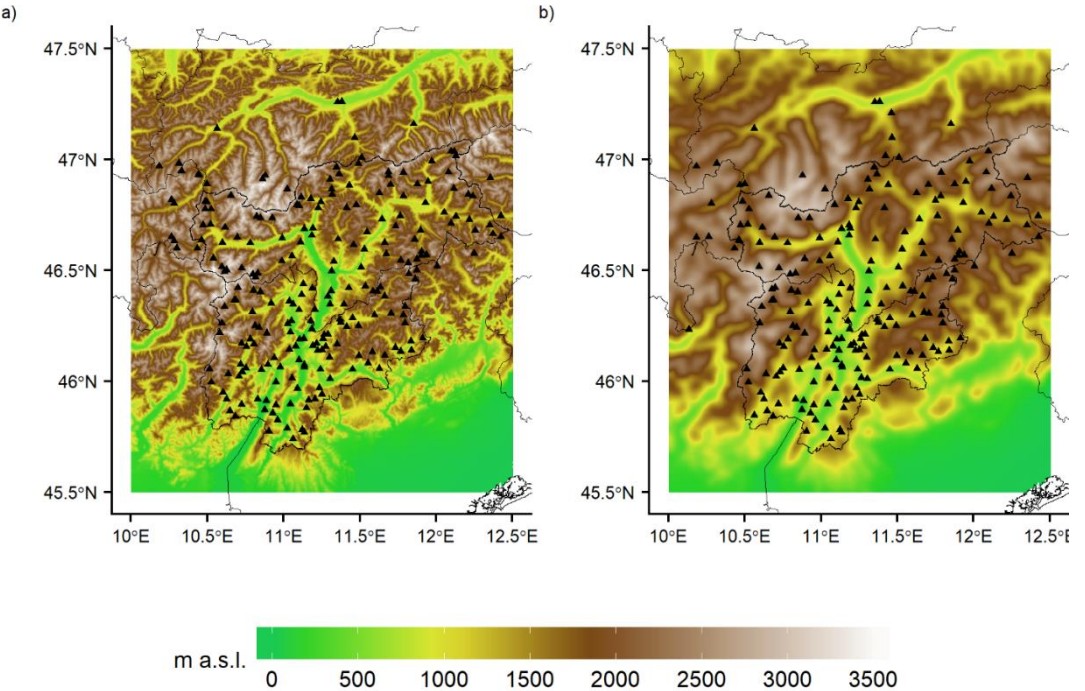

**Figure 2: Spatial distribution of a) temperature and b) precipitation weather stations superimposed to the topography considered in the interpolation scheme. In panel a) the digital resolution model (DEM) at 250 m resolution is reported, in panel b) the smoothed version of the 250 m DEM used for precipitation spatialization is shown.**

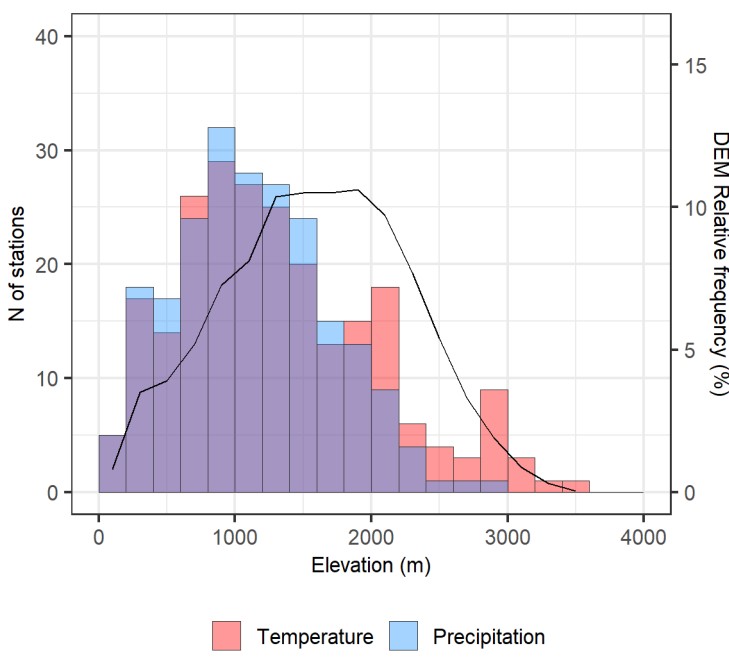

**Figure 3: Elevation distribution of temperature and precipitation stations in absolute numbers over 200 m bins (left axis). The relative elevation distribution of the 250 m digital elevation model covering the study domain is reported on the secondary y-axis for the same 200 m bins.**

The data availability also varies over time (Fig. 4). The first relevant improvement of data coverage occurs during the 1970s, while the greatest increase is observed after 1980 and, especially for temperature, after 1990 when the automatic stations started to operate in most areas. Due to the significantly lower station availability before the 1980s, which could reduce the general accuracy of the results for the early decades, the starting year for the computation of the gridded dataset for both temperature and precipitation was set to 1980. Although the coverage of temperature series is less dense than that of precipitation data before 1990, this it is not expected to affect the general robustness of the results thanks to the greater spatial coherence typically shown by temperature records (Brunetti et al., 2006). The effects of the variability in data availability on the result accuracy was discussed more in detail in Sect. 3.1.

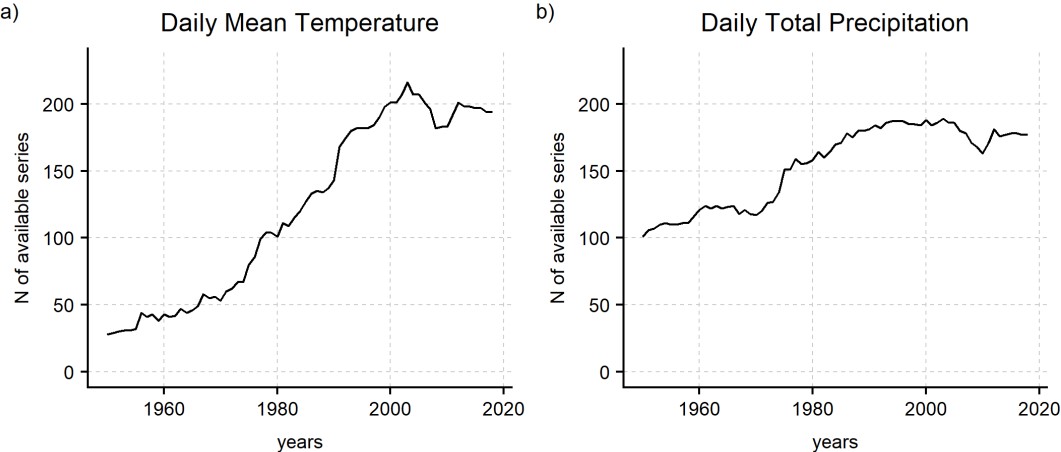

 **Figure 4: Number of available stations over 1950 – 2018 period in the collected databases of a) temperature and b) precipitation.**

**2.3 The interpolation scheme**

The 250 m resolution fields of daily mean temperature and daily precipitation over Trentino – South Tyrol were computed over the period 1980 – 2018 by applying the anomaly-based approach. In this framework, the final gridded output is derived from the superimposition of the gridded station daily anomalies and the interpolated climatologies, i.e. 30 year means, for the

reference period 1981 – 2010. The interpolation grid and all topographic features were derived from the Digital Elevation Model (DEM) Copernicus EU-DEM v1.1 (https://land.copernicus.eu/imagery-in-situ/eu-dem/eu-dem-v1.1), which was aggregated from the original 25 m to the target 250 m resolution. The elevation values of the DEM used for precipitation interpolation were further smoothed by replacing the elevation of each point by a weighted average of the surrounding grid cell elevations with weights halving at 2 km distance from the considered cell. The halving distance was chosen as the one

minimizing the modelling errors (see Sect. 3.1 for a description of errors). As suggested in previous work, this procedure can avoid entering in the interpolation too fine terrain details with respect to the expected scales of interaction between atmospheric circulation and orography (Brunetti et al., 2012; Foresti et al., 2018). In the following the two steps of the interpolation procedure are described, separately.

**2.3.1 The 1981 – 2010 climatologies**

In order to derive the 1981 – 2010 climatological fields of mean temperature and precipitation, the daily series were aggregated at monthly scale and the monthly climatological normals, defined as the averages over the 30 year period, were computed for each station in the database. In order to prevent the station normals from being biased due to the different number of available monthly data over 1981 – 2010, before their calculation, all missing monthly values in the reference interval for each series were reconstructed by means of the nearby stations (Brunetti et al., 2014; Crespi et al., 2018).

The interpolation of the monthly station normals on the 250 m resolution grid was based for both temperature and precipitation on the PRISM scheme developed by Daly et al. (2002). More specifically, the monthly climatological values at each target grid point were computed by applying a linear relation with elevation which was estimated for each month from the surrounding stations by means of a weighted linear regression:

$$z_m(x,y) = \alpha_m(x,y) + \beta_m(x,y) \cdot h(x,y) \qquad (1)$$

where $z_m(x,y)$ is the climatology of month $m$ at the target point $(x,y)$, $\alpha_m(x,y)$ and $\beta_m(x,y)$ are the local regression coefficients for month $m$ and $h(x,y)$ is the grid point elevation. All stations were weighted in order to select and assign a greater contribution in the weighted linear fit to the sites which were closer to the target point and with the most similar physiographic features. For both temperature and precipitation interpolation, the geographical features considered for the station weight were the horizontal distance and the difference in elevation and slope conditions, i.e. steepness and orientation,

with the target grid point. Other potential geographical features which can have an influence on the modelling of the climate spatial gradients, e.g. the difference in sea distance, were not included since their contribution to the total weight was found to be negligible.

The weight of each station $i$ was then expressed for each month $m$ and grid point $(x,y)$ as the product of the single weighting functions for each of the considered geographical parameter $k$:

$$w_{i,m}(x,y) = \prod w_{i,m}^k(x,y) \qquad (2)$$

where $k$ is the geographical feature, i.e. horizontal distance, elevation difference, difference in slope steepness and slope orientation. The single weighting function was defined as a Gaussian function as follows:

$$w_{i,m}^k(x,y) = e^{-\left(\frac{\left(\Delta_{i,m}^k(x,y)\right)^2}{c_m^k}\right)} \qquad (3)$$

where $\Delta_{i,m}^k$ is the difference in the geographical parameter between the station $i$ and the grid point, and $c_m^k$ is the corresponding

decay coefficient which was kept constant over the entire domain and estimated for each month and geographical feature by means of an optimization procedure. The weighted linear fit at each target point was performed by selecting the 35 stations, for temperature, and the 15 stations, for precipitation, with the highest weight. Also the number of stations entering in the fit was defined by an optimization procedure minimizing the model errors.

### 2.3.2 The daily anomalies and the absolute fields

The fields of daily mean temperature and daily precipitation anomalies were computed over the period 1980 – 2018 for the 250 m resolution grid. To this aim, the station daily records were converted into series of daily anomalies. More precisely, for mean temperature, the daily normals of each station were firstly obtained by interpolating the corresponding monthly normals by means of the first two harmonics of a Fourier series and the daily anomalies were then computed as the difference between the daily observed temperature and the daily normal for the corresponding calendar day. As regards precipitation, daily

anomalies were defined as the direct ratio between the daily precipitation record and the climatological value of the corresponding month.

The station daily anomalies were interpolated onto the grid through an IDW-based scheme:

$$a_t(x,y) = \frac{\sum_{i=1}^{N} w_i(x,y) \cdot a_{i,t}}{\sum_{i=1}^{N} w_i(x,y)} \tag{4}$$

where $a_t(x,y)$ is the daily anomaly of mean temperature or precipitation at the target point $(x,y)$ for the daily step $t$, $a_{i,t}$ is the anomaly of the station $i$ for the date $t$, $w_i(x,y)$ is the weight of station $i$ relative to the grid point $(x,y)$ and $N$ is the total number of station records available for the date $t$. As for the climatology interpolation, the station weight was expressed as in Eq. (2) and Eq. (3), but in this case it depended only on horizontal and vertical distances. In the case of precipitation, in order to reduce the simulation of false wet days, the daily estimate was set to zero if no precipitation was recorded at the three closest stations on the day under reconstruction.

Finally, the gridded anomalies were combined with the 1981 – 2010 climatologies to derive the daily fields of mean temperature and precipitation in absolute values as follows:

$$T_t(x,y) = a_t(x,y) + \bar{T}_d(x,y) \tag{5}$$

$$P_t(x,y) = a_t(x,y) \cdot \bar{P}_m(x,y) \tag{6}$$

where $T_t(x,y)$ and $P_t(x,y)$ are the interpolated values at the grid point $(x,y)$ on date $t$ of mean temperature and precipitation, respectively, $\bar{T}_d(x,y)$ is the daily temperature climatology for the corresponding calendar day and $\bar{P}_m(x,y)$ is the monthly precipitation climatology for the corresponding calendar month. The absolute daily precipitation fields were derived directly as the product of daily anomalies and the corresponding monthly climatologies obtained by Eq. (1). In the case of temperature, in order to better account for the annual seasonality, the monthly normals for each grid point from Eq. (1) were firstly fitted by means of the same second-order trigonometric function used at station level to get a reference value $\bar{T}_d$ for each day of the year. The absolute temperature estimates were then computed as in Eq. (6) by adding the daily anomalies to the fitted climatological values on the corresponding calendar day.

## 3 Results and Discussion

### 3.1 The dataset validation

The uncertainty evaluation of the gridded datasets is essential to properly apply the products and interpret the results. One of the most important aspects to consider is that the grid-cell values obtained by the spatial interpolation represent areal mean estimates of temperature and precipitation. The punctual conditions at single station sites, especially the daily precipitation peaks, result to be smoothed after the spatialization so that the fine resolution of the daily grids does not correspond to the scales effectively resolved, which are limited by the horizontal spacing of the station network.

The accuracy of the gridded dataset of daily mean temperature and precipitation was evaluated by applying the anomaly-based reconstruction scheme to simulate the daily records of all stations in Trentino – South Tyrol over the study period 1980 – 2018

in a leave-one-out approach, i.e. by removing the station data under evaluation in order to avoid self-influence. The same approach was also applied to assess the errors of the interpolated 1981 – 2010 monthly climatologies, separately. The reconstruction accuracy was computed by comparing the simulations and observations in terms of mean error (BIAS), mean absolute error (MAE), root mean square error (RMSE) and correlation. For daily temperature reconstruction, the resulting

BIAS, as average over all stations and whole time period, was almost zero and a MAE (RMSE) of around 1.5 °C (1.9 °C) was obtained. The agreement between daily observations and simulations was high with a mean Pearson correlation coefficient of 0.97 (Fig. 5). As regards the extremes, the correlation was still high if only the temperature values above 95th percentiles were considered with a mean Pearson coefficient of 0.94, while a slight overestimation was observed for the simulation of temperature records below the 5th percentile (Fig. 5). The reconstruction errors and correlation by months were also evaluated

(Table 1). MAE ranges from 1.1 °C in July to 1.8 °C in October, when the lowest correlation (0.80) was also obtained. BIAS is within -0.5 – 0.5 °C in all months. The simulated 1981 – 2010 monthly mean temperature climatologies at all station sites in the study region showed zero BIAS and MAE (RMSE) ranging from 0.5 (0.6) °C in May to 0.9 (1.1) °C in January (Table 2).

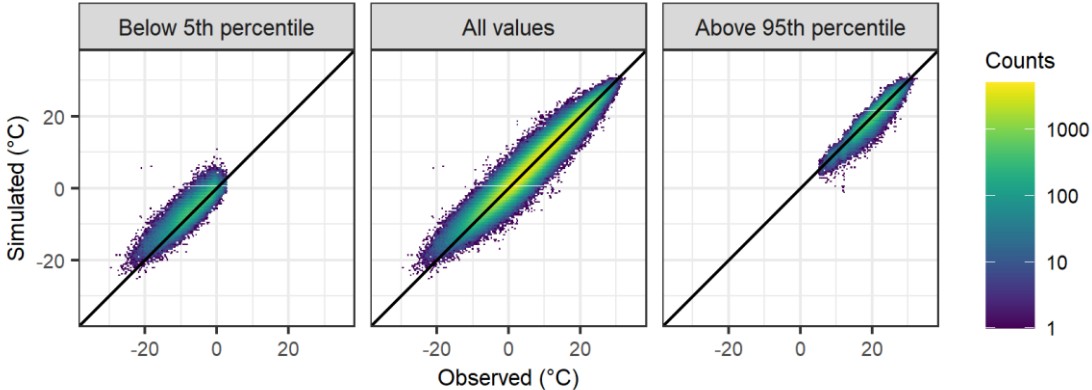

**Figure 5: Distribution of simulated and observed daily mean temperature values for all station sites in the study region over the period 1980 – 2018. The three panels report the comparison by considering only values below the 5th percentile, all values and only values above the 95th percentile.**

| MONTH | Daily mean temperature | | | | Daily precipitation | | | |
|---|---|---|---|---|---|---|---|---|
| | BIAS (°C) | MAE (°C) | RMSE (°C) | CORR | BIAS (mm) | MAE (mm) | RMSE (mm) | CORR |
| 1 | 0.1 | 1.5 | 1.9 | 0.88 | 0.0 | 0.6 | 2.0 | 0.91 |
| 2 | -0.2 | 1.5 | 1.9 | 0.90 | 0.0 | 0.6 | 2.1 | 0.91 |
| 3 | 0.5 | 1.5 | 1.9 | 0.89 | 0.0 | 0.7 | 2.3 | 0.90 |
| 4 | -0.1 | 1.5 | 1.8 | 0.86 | 0.0 | 1.0 | 2.6 | 0.92 |
| 5 | 0.4 | 1.5 | 1.9 | 0.85 | 0.0 | 1.4 | 3.2 | 0.91 |
| 6 | 0.1 | 1.3 | 1.6 | 0.90 | 0.0 | 1.6 | 3.6 | 0.88 |
| 7 | 0.3 | 1.1 | 1.4 | 0.91 | 0.0 | 1.7 | 4.1 | 0.86 |

| | | | | | | | | |
|---|---|---|---|---|---|---|---|---|
| 8 | 0.3 | 1.1 | 1.4 | 0.90 | 0.0 | 1.6 | 3.9 | 0.87 |
| 9 | -0.4 | 1.5 | 1.9 | 0.82 | 0.0 | 1.1 | 3.3 | 0.92 |
| 10 | 0.4 | 1.8 | 2.2 | 0.80 | 0.0 | 1.1 | 3.5 | 0.94 |
| 11 | -0.2 | 1.7 | 2.1 | 0.84 | 0.0 | 1.1 | 3.3 | 0.94 |
| 12 | -0.3 | 1.6 | 2.0 | 0.87 | 0.0 | 0.7 | 2.4 | 0.92 |

**Table 1: Monthly mean leave-one-out reconstruction errors and correlation for Trentino – South Tyrol daily mean temperature and daily precipitation series over the period 1980 – 2018. BIAS is computed as difference between simulations and observations.**

For precipitation, the reconstruction errors by averaging over all stations and daily records showed a MAE (RMSE) of 1.1 (3.2) mm, zero BIAS and a mean correlation coefficient of 0.91. By considering wet days only, i.e. daily precipitation values ≥ 1 mm, the mean correlation decreased to 0.87 and errors increased with mean BIAS of -0.6 mm and MAE (RMSE) of 3.4 (5.8) mm, suggesting a tendency to underestimate the higher precipitation records. The mean monthly errors ranged from 0.6 in January to 1.7 mm in July for MAE and from 2.0 mm in January to 4.1 mm in July for RMSE (Table 1). As regards the precipitation climatological reconstruction, BIAS is below 0.5 mm in all months and MAE (RMSE) ranges from 4.1 (5.6) mm in February to 10.7 (14.2) mm in November (Table 2). It is worth noting that systematic errors in rain-gauge measurements can derive from the instrument type, site elevation and exposition to wind. For instance, a well-known source of uncertainty affecting precipitation records, especially in mountain environment, is the "rain-gauge undercatch" which could be particularly relevant during episodes of strong wind and solid precipitation and could account up to several tens of percent of the measured values (Frei and Schär, 1998: Sevruk et al., 2009). Several approaches were developed to account for the undercatch at station sites (see e.g. Grossi et al., 2017), however the magnitude of correction is highly variable with locations, sensors and seasons and unproper corrections could reduce the general accuracy of the dataset. For this reason, the contribution of rain-gauge undercatch was not quantified in the current analyses.

| MONTH | monthly mean temperature climatologies | | | monthly precipitation climatologies | | |
|---|---|---|---|---|---|---|
| | BIAS (°C) | MAE (°C) | RMSE (°C) | BIAS (mm) | MAE (mm) | RMSE (mm) |
| 1 | 0.0 | 0.9 | 1.1 | 0.1 | 5.2 | 7.1 |
| 2 | 0.0 | 0.7 | 0.9 | 0.2 | 4.1 | 5.6 |
| 3 | 0.0 | 0.6 | 0.7 | 0.3 | 5.5 | 7.5 |
| 4 | 0.0 | 0.5 | 0.6 | 0.2 | 8.2 | 11.8 |
| 5 | 0.0 | 0.5 | 0.6 | -0.1 | 8.4 | 11.5 |
| 6 | 0.0 | 0.5 | 0.7 | -0.1 | 7.9 | 10.3 |
| 7 | 0.0 | 0.5 | 0.7 | 0.1 | 7.4 | 9.6 |
| 8 | 0.0 | 0.5 | 0.7 | 0.3 | 7.1 | 9.3 |
| 9 | 0.0 | 0.5 | 0.7 | 0.1 | 7.4 | 9.6 |
| 10 | 0.0 | 0.6 | 0.7 | 0.2 | 9.2 | 12.3 |
| 11 | 0.0 | 0.7 | 0.8 | 0.3 | 10.7 | 14.2 |
| 12 | 0.0 | 0.8 | 1.0 | 0.1 | 7.2 | 10.1 |

**Table 2: Monthly mean leave-one-out reconstruction errors of the 1981 – 2010 monthly mean temperature and precipitation climatologies for Trentino – South Tyrol sites. BIAS is computed as difference between simulations and observations.**

Due to the high variability of interpolation errors with the daily precipitation intensity, the comparison between simulated and observed precipitation in wet days during winter and summer was assessed by splitting the data for intensity intervals (Fig. 6).

Figure 6 confirms the general tendency of the daily reconstruction method to underestimate intense daily precipitation totals, especially in summer when the underestimation in simulations for the highest quantile interval (0.98 – 0.999) is in median about 24%. On the contrary, overestimation of the lowest quantiles (0.1 – 0.2) in summer was depicted with a median exceedance of 30%. The greater difficulty in simulating summer precipitation could be mostly ascribed to the higher spatial variability of summer precipitation events, mainly driven by convection.

In order to assess also the temporal variability of the dataset uncertainty over the study period and the influence of changes in data coverage over time, the daily simulations and observations were aggregated at monthly scale and reconstruction errors were evaluated for each month over all available stations. The annual averages of monthly errors over 1980 – 2018 are reported in Fig. 7. It is worth noting that the values are almost stable over the whole period for both temperature and precipitation with MAE around 0.7 °C and 13 mm, respectively. However, temperature errors slightly decreased after 1990, as a consequence of

the increase in station density (see Fig. 4a). As regards precipitation, a greater error variability can be observed after 2000 probably due to the changes occurred in station networks during the transition from the manual to automatic rain gauges. The resulting reconstruction errors and their limited variability over the spanned period enable the application of the derived dataset for the assessment of climatological trends in temperature and precipitation. Such applications could be useful to compare and integrate previous existing studies analysing long-term climate trends and their spatial patterns over the region (e.g. Brugnara

et al. 2012). A trend analysis was beyond the scope of the current study, and thus not included in the present work but it will be discussed in forthcoming studies.

a)

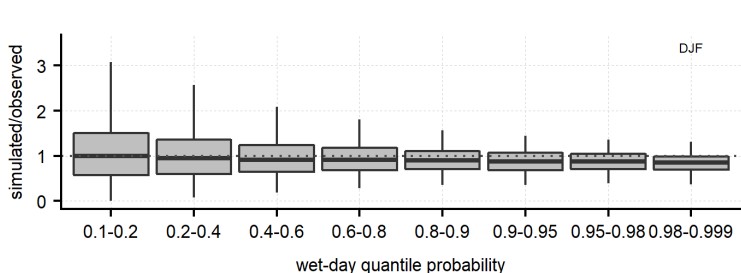

b)

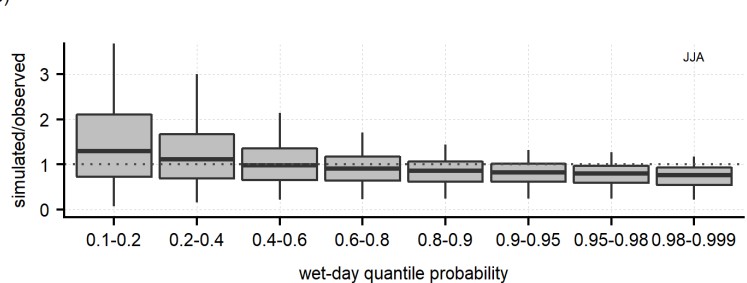

**Figure 6: Ratio between daily simulated and observed precipitation (wet days only) in a) winter and b) summer over 1980 – 2010 grouped for quantiles. Boxplots extend over the interquartile range with the median reported by the bold line, while whiskers extend over the full range of outliers.**

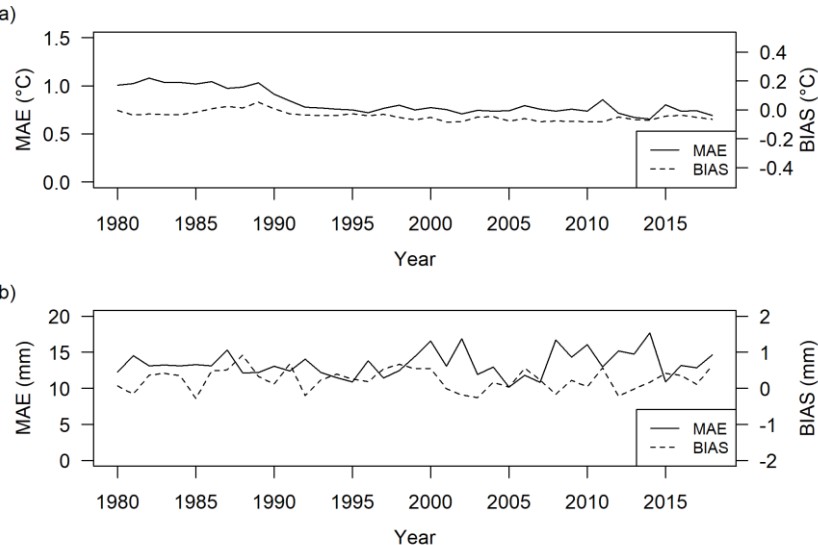

**Figure 7: Annual series of mean monthly leave-one-out reconstruction errors (MAE and BIAS) of a) temperature and b) precipitation over the study period for the available stations in Trentino – South Tyrol.**

### 3.2 The gridded dataset: regional climatic features, example cases and snow-cover comparison

The 250 m gridded dataset allows to discuss and analyse the main features of the climate in the region. In the following, the 30 year annual climatologies of mean temperature and total precipitation over the period 1981 – 2010 are shown in Fig. 8. The derived features were largely in agreement with the findings of previous works focusing on the regional climate (e.g. Adler et al., 2015), and the 250 m grid spacing allows a detailed visualization of the spatial patterns. The mean annual temperature ranges from +14 °C in the Garda Valley located in the southernmost part of the region to about -11 °C at the Ortles peak with an average value of around +5 °C over the entire study region. The thermal contrast between the inner valleys and the steep surrounding reliefs is well depicted. Mean annual values are around +12 °C in the main valley bottoms, while the isothermal of 0 °C occurs at about 2400 m. The annual cycle of mean temperature, as average over the region and based on the 1981 – 2010 normals, is characterized by the warmest conditions in July while the coldest month is January with -3.2 °C (Fig. 9). The greatest warming occurs between April and May, when also the thermal range between the coldest and the hottest locations increases reaching the maximum between May and June with almost 30 °C. The most relevant cooling is depicted in the transition from October to November when the isothermal of 0 °C drops from around 2700 to 1700 m and the occurrence of cold air pools in the valleys becomes more frequent.

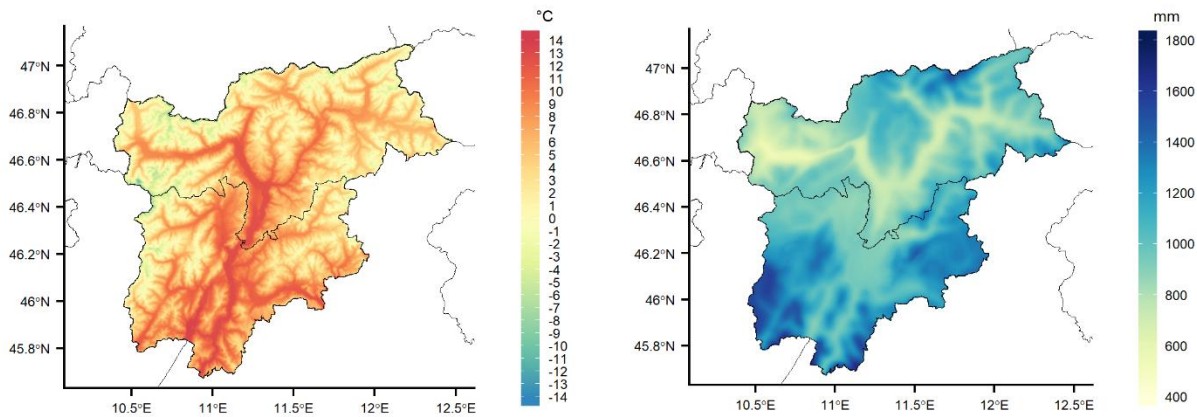

**Figure 8: 1981 – 2010 annual a) mean temperature and b) total precipitation climatologies (250 m grid spacing) over Trentino – South Tyrol.**

Topographic effects are also evident in the precipitation distribution over the study region. Annual totals range from 530 mm in the inner Venosta Valley (north-western South Tyrol) to more than 1700 mm in the southernmost part of the region, where wet southerly flows are more relevant. The mean annual precipitation sum is around 1000 mm as spatial average over the whole region, with drier conditions in the northern inner valleys of South Tyrol with annual totals around 900 mm. As regards the mean annual cycle, the driest conditions occur in late winter, with a minimum in February when the mean regional totals are below 40 mm. On the contrary, precipitation significantly increases in May, with the growing contribution of local convection, and the annual maximum is reached in July, with a regional average of almost 120 mm. Regional mean precipitation values show a secondary minimum in September and increase again in October, when greater contrasts over the region, especially between north and south, are depicted with the wettest contributions in the Trentino valleys, more exposed to humid air masses coming from the Mediterranean sea .

By focusing on distinct portions of the study area, a relevant sub-regional variability can be observed (Fig. 10). In particular, a different annual precipitation cycle characterizes the northern and southern portions of the study region. In the northern and central parts the precipitation cycle has a maximum in summer and the peak increases moving from west (Venosta Valley), where rain-shadow effects are more frequent, to east, where the Alpine ridge receives wet contributions from both northerly and southerly flows. On the contrary, the annual cycle in the south of the region shows two precipitation maxima occurring in spring and autumn. The influence of subtropical high-pressure areas is particularly relevant for the southernmost valleys and contributes to the drying tendencies over summer months which defines here the local precipitation minimum.

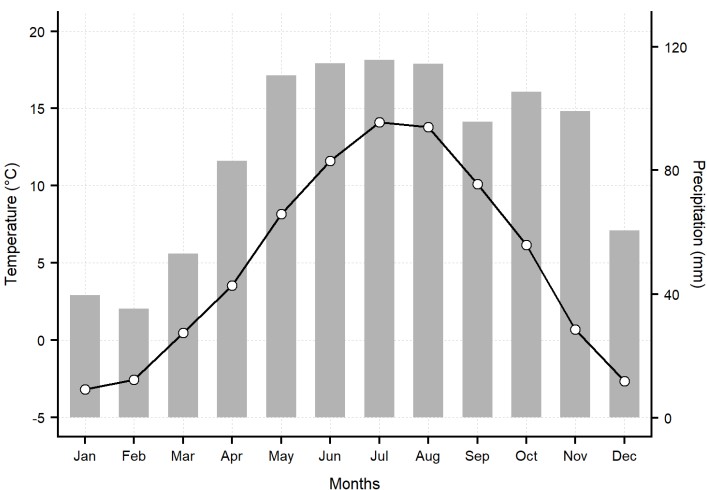

**Figure 9: Mean regional annual cycle of monthly mean temperature and total precipitation based on the 1981 – 2010 climatologies.**

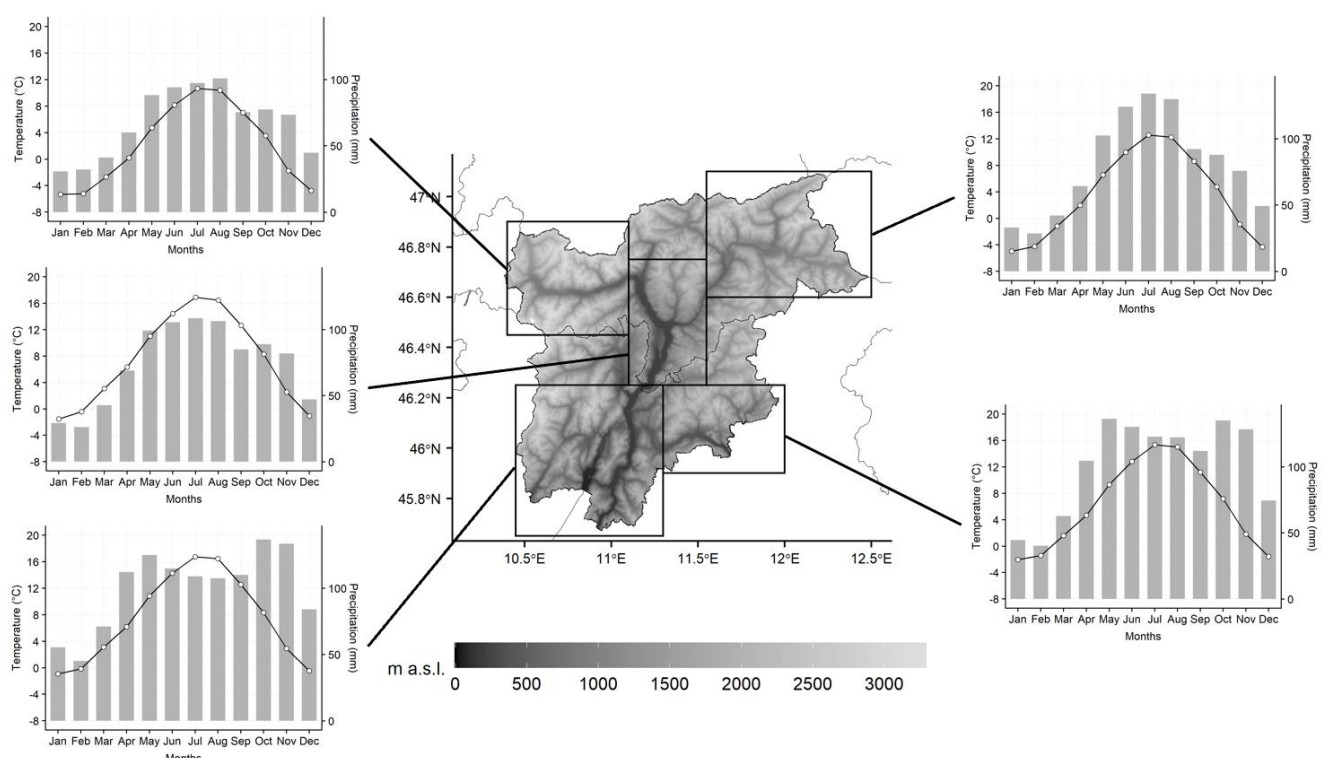

**Figure 10: Mean sub-regional annual cycles of monthly mean temperature and total precipitation based on the 1981 – 2010 climatologies for five distinct areas.**

Among the useful applications of high-resolution gridded climate variables at daily timescale, they allow to study and reconstruct past episodes of particular interest, such as intense precipitation events, and their temporal day-by-day pattern over

the entire region. In particular, we reported as example cases the computed 250 m daily fields for two past episodes experiencing meteorological extremes over the region.

    One of the most recent intense events occurred on 27–30 October 2018 when an intense storm affected a large portion of eastern Italian Alps leading to severe impacts, such as floods, landslides, interruption of traffic and electricity supply and severe forest damages due to intense wind gusts up to 200 km h$^{-1}$ (Dalponte et al., 2020; Davolio et al., 2020). During this

event, exceptional amounts of precipitation were recorded in few days at many locations. In Fig. 11, the 4 day precipitation sum over 27–30 October 2018 is reported together with the daily anomalies with respect to the 1981 – 2010 monthly normals for the month. In some areas, especially the upper Isarco Valley in the northern of South Tyrol, the south western portion of Trentino and along its eastern border with Veneto region, precipitation amounts were particularly intense, with 4 day totals locally exceeding 500 mm (Fig. 11a). Over the whole region, the 4 day precipitation totals were greater than the monthly

normals for October, with 80% of the grid points with precipitation amounts more than double the climatological totals for the month (Fig. 11b). As discussed in Sect. 3.1, the reported daily fields provide a comprehensive overview of the precipitation patterns over the region and the locations of the maxima, however the applied daily interpolation based on weighted spatial average has a smoothing effect so that the daily gridded fields are expected to underestimate the very localized peaks at individual stations.

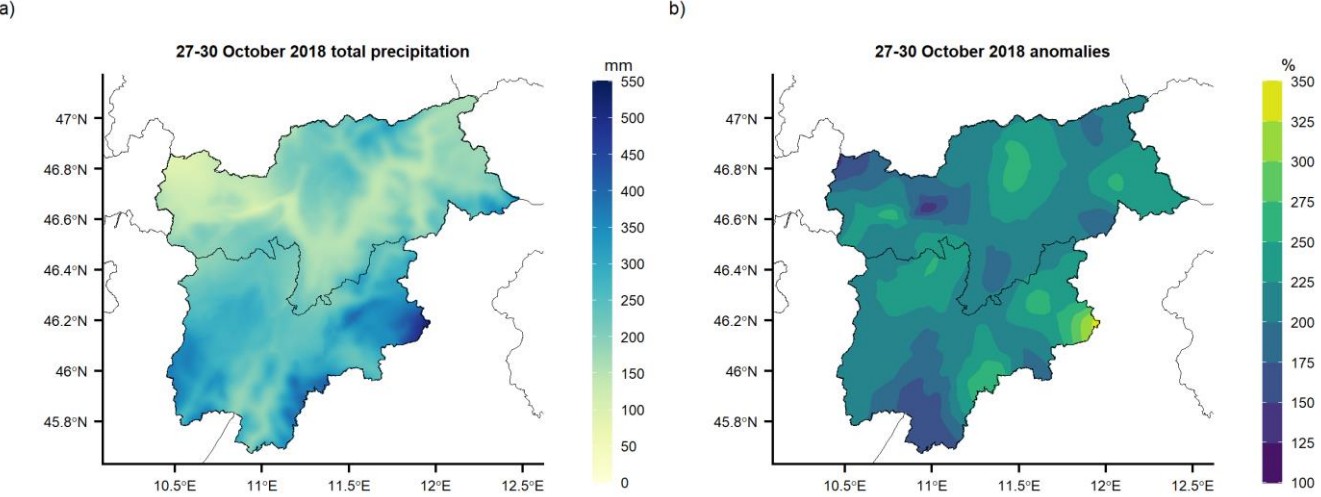

**Figure 11: Event of October 2018 over Trentino – South Tyrol: a) 4 day total precipitation over 27–30 October 2018 (250 m spatial resolution) and b) relative anomalies of the 4 day totals with respect to the 1981 – 2010 monthly normals of total precipitation for October.**

As second example case, the summer 2003 was considered when exceptionally high temperature values were recorded in a large part of western Europe due to the influence of sub-tropical air masses and the presence of a wide high-pressure system over Europe. The high temperatures and the relatively scarce precipitation occurred in the previous winter and spring months fostered drought conditions in several regions. In Trentino – South Tyrol the mean temperature anomalies over the summer

period were, as average over the region, around +3 °C with respect to the reference 1981 - 2010 normals (Fig 12a). In particular,

August was the hottest month and the greatest temperature values were reached on 11 August with a mean temperature on regional level greater than 21 °C and daily maximum temperature close to +40 °C in Bolzano and Trento. The distribution of the daily mean temperature on 11 August shows values above +30 °C along the main valley bottoms (Fig. 12b) and over almost the whole region the daily values exceed the climatological means with a mean anomaly of about +7 °C for that day.

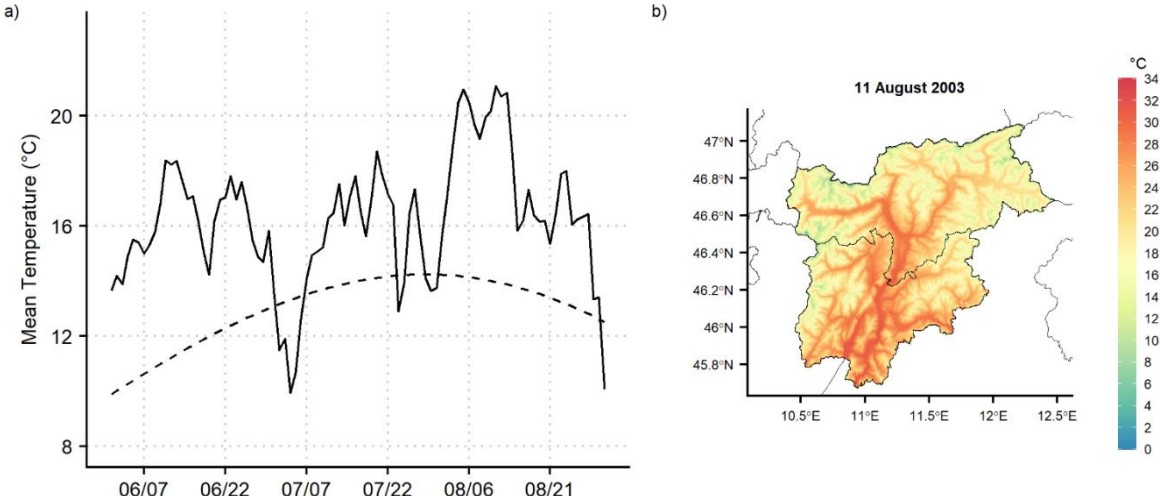

**Figure 12: in panel a) the regional daily mean temperature series for summer 2003 is shown (solid line) together with the 1981 –**
**2010 daily normals (dashed line); in panel b) the spatial distribution over Trentino – South Tyrol of mean temperature on the hottest summer day in 2003 (11 August) is reported at 250 m grid spacing.**

In order to provide an example of application of the gridded meteorological data in combination with other high-resolution products, the dataset of mean temperature and precipitation was compared to a 250 m resolution daily dataset of snow cover available over the region. The daily snow-cover maps used for the comparison are based on Moderate Resolution Imaging

Spectroradiometer (MODIS) images and provide binary snow-cover information (snow, snow free) using an algorithm tailored to complex terrain (Notarnicola et al., 2013). In this product, clouds were interpolated in Terra-only images yielding almost cloud-free daily maps for the period 2000 – 2018 (Matiu et al., 2020). Starting from the snow-cover data, the snow-cover duration (SCD) over the winter season (December, January and February) was computed at each grid point of the study domain and the Pearson correlation with the mean winter temperature and total winter precipitation was computed over all available

winters of SCD data (2001 – 2018). Since SCD saturates at high elevations preventing from computing the correlation coefficients, all points above 2500 m were excluded from the analysis. As expected, correlation coefficients with temperature are negative within the inter-quantile range ($25^{th}$ – $75^{th}$ percentiles) for all elevation intervals (Table 3). The anticorrelation decreases with increasing elevation with median values ranging from -0.55 below 500 m to -0.41 above 1500 m. On the contrary, the correlation with total winter precipitation is positive within the inter-quantile range in all cases and a clear

elevation dependency is evident with correlation values increasing with altitudes, up to 0.55 as median in the range 1500–1750 m. In considering the behaviour of SCD correlation with temperature and precipitation over different elevation bands, the

effect of winter SCD saturation with increasing elevation should be taken into account. This could partly explain the increase of the anticorrelation with temperature and the concurrent decrease of precipitation–SCD dependency in the upper elevation bands (above 1750 m). These outcomes are in agreement with other literature studies investigating the dependency of several
snow parameters with temperature and elevation in the Alps (see e.g. Móran-Tejeda et al., 2014; Schöner et al., 2019; Matiu et al., 2021) and specifically in Trentino region (Marcolini et al., 2017).

| Elevation interval (m) | SCD and $T_m$ correlation | | | SCD and P correlation | | |
|---|---|---|---|---|---|---|
| | 25th percentile | Median | 75th percentile | 25th percentile | Median | 75th percentile |
| < 500 | -0.63 | -0.55 | -0.44 | 0.01 | 0.14 | 0.28 |
| 500 - 750 | -0.60 | -0.51 | -0.38 | 0.14 | 0.28 | 0.43 |
| 750 - 1000 | -0.60 | -0.49 | -0.33 | 0.25 | 0.39 | 0.52 |
| 1000 - 1250 | -0.58 | -0.45 | -0.27 | 0.34 | 0.47 | 0.58 |
| 1250 - 1500 | -0.56 | -0.42 | -0.25 | 0.40 | 0.52 | 0.64 |
| 1500 - 1750 | -0.52 | -0.41 | -0.26 | 0.43 | 0.55 | 0.66 |
| 1750 - 2000 | -0.51 | -0.43 | -0.34 | 0.34 | 0.43 | 0.56 |
| 2000 - 2500 | -0.50 | -0.44 | -0.37 | 0.10 | 0.25 | 0.36 |

Table 3: Median and inter-quantile range of 2001 – 2018 correlation coefficients over different elevation bands between SCD (snow-cover duration) and a) mean temperature and b) total precipitation in winter season (December, January and February). Only grid points below 2500 m are considered.

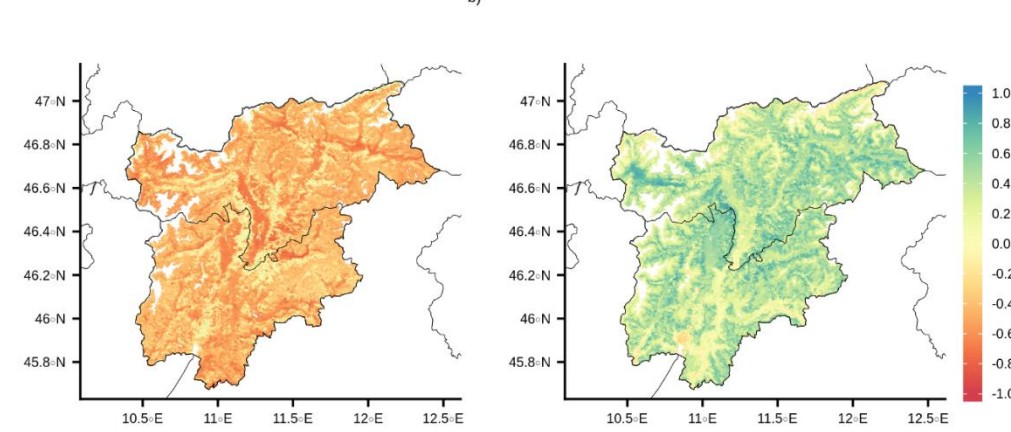

Figure 13: Spatial distribution of 2001 – 2018 correlation coefficients over the region between SCD (snow-cover duration) and a) mean temperature and b) total precipitation in winter season (December, January and February). White areas correspond to masked grid points (above 2500 m) or to missing snow cover data.

The elevation dependency of winter SCD correlation with temperature and precipitation is also highlighted in the spatial distribution of the coefficients over the region (Fig. 13). The greatest anticorrelation with temperature is depicted along the main valley floors, especially Adige and Isarco Valleys, while the greatest dependency with precipitation is pointed out along
the mid-elevation zones decreasing towards both lower elevations, where winter precipitation occurs generally in liquid form,

and higher altitudes where snowfalls usually occur in winter season and, regardless of their magnitude, snow lasts on the ground due to the relatively low temperature.

## 4 Codes and data availability

The dataset of daily mean temperature and total precipitation at 250 m resolution spanning the period 1980 – 2018 for Trentino
– South Tyrol and the 250 m resolution 1981 – 2010 monthly climatologies are freely available at PANGAEA Data Publisher for Earth and Environmental Science through https://doi.org/10.1594/PANGAEA.924502, Crespi et al. (2020). The data are stored in NetCDF format that eases the processing in scientific programming software (e.g. Python and R) and GIS. The dataset of daily snow cover used for the comparison is freely available from Matiu et al. (2019). All routines are developed in R environment and are available upon request from the authors.

**5 Conclusions**

The 250 m resolution dataset of daily mean temperature and total precipitation for Trentino – South Tyrol region over 1980 – 2018 was presented. It was derived starting from a dense database of more than 200 station observations, covering the region and close surroundings, which were checked for quality and homogeneity. The gridded daily fields were computed by applying an interpolation procedure in which the 30 year climatologies (1981 – 2010) and the daily anomalies were combined. In this
scheme, the local relationship between the climate variables and the orography are considered for the climatology spatialization, while the station daily anomalies are interpolated by a weighted-averaging approach. The method was demonstrated in previous studies to be particularly robust in reconstructing climate fields in mountain areas, where systematic errors, especially precipitation underestimations, could occur due to the uneven data coverage, e.g. between low and high-elevation areas.
The leave-one-out cross-validation pointed out the overall robustness of gridded fields for both daily temperature and precipitation with mean correlation coefficients above 0.80 in all months and MAE, as average over all stations and months, of around 1.5 °C for mean temperature and 1.1 mm for precipitation. Moreover, the reconstruction errors showed to be almost constant over the whole study period, despite the changes in available station number, especially during the first decade.
The provided dataset represents a valuable source of continuous climate information for the region, spanning almost 40 years,
and the very fine grid spacing facilitates its application for a wide range of scopes requiring spatially explicit fields of climate variables, such as hydrological analyses, environmental modelling, impact studies and remote sensing data validation. The availability of gridded information allows to derive more accurate aggregations and spatial averages over specific sub-domains, e.g. at catchment and sub-catchment levels, than averaging the single station records directly. In addition, the dataset could represent an archive of climate information to support the assessment of spatio-temporal variability and trends in temperature
and precipitation over the region and integrate previous existing studies. The suitability of the dataset for such analyses is

supported by the use of quality-controlled station observations, the application of homogenization procedures, the results of the accuracy evaluation obtained for the entire spanned period and the regular updates. Other potential applications of the dataset include the downscaling of climate change scenarios and the improved evaluations of high-resolution regional climate models.

However, it is necessary to take into account for any application that a larger uncertainty should be associated to the gridded estimates for points at higher altitudes due to the decrease of data coverage with increasing elevation. In addition, the spatial scales effectively resolved by the dataset at daily resolution are coarser than the nominal grid spacing and in the order of several km, i.e. close to the mean inter-station distance, and systematic errors in punctual values cannot be avoided, such as underestimation of the highest intensities. Possible improvements in spatial representativeness could be derived by the

integration of additional in situ observations, especially at higher elevations, and/or by evaluating alternative and more sophisticated interpolation techniques for modelling the small-scale interactions between climate variables and orography. In this context, an extended inter-comparison work with existing products covering the region at different temporal and spatial scales would be useful to better investigate the benefits and limitations of the dataset and it will be addressed to a forthcoming study.

Despite the intrinsic issues, the dataset allows to extract and analyse the fine scale distribution of the main climatic features over the region, which are described by the 1981 – 2010 monthly climatologies. The suitability of the daily fields in representing the spatial structure of specific events of interest was discussed and example cases of intense past episodes were reported. In order to provide an example of the potential applications of this fine-scale product with other high-resolution data, we also performed a comparison between the gridded meteorological variables and the 250 m MODIS SCD winter maps. The

pointwise correlation analysis over the period 2001 – 2018 pointed out a clear elevation dependency of SCD to both temperature and precipitation and the fine spatial scale led to a more detailed insight on the spatial distribution over the region of these relationships suggesting interesting open questions which require to be further developed in forthcoming works.

The dataset is intended to be integrated in the next future by the gridded fields of daily maximum and minimum temperature and to be converted into a near-real time product including regular updates and supporting operational applications at local

and regional levels.

**Author contributions**

AC implemented and performed the data interpolation. The data collection and processing were performed by AC in collaboration with MM and GB. The validation analyses were carried out by AC, MP contributed in the discussion of the interpolation errors and MM provided snow cover data and supported the interpretation of the results. MZ supervised and

supported all phases of the work. AC prepared the manuscript draft and the revised version with assistance from MM. AC, MM, BG, MP and MZ collaborated to the editing.

**Competing interests**

The authors declare that they have no conflict of interest.

**Acknowledgments**

The authors thank the Hydrographic Office of the Autonomous Province of Bolzano and Meteotrentino for the provision of the meteorological data for the study region and ARPA Veneto, ZAMG, the HISTALP project and MeteoSwiss which provided the series used for the extra regional sites. Prof. Majone and Prof. Bellini from the University of Trento are also acknowledged for providing useful information on available data and products. The research leading to these results has received funding from the European Regional Development Fund, Operational Programme Investment for growth and jobs ERDF 2014-2020 500 under Project number ERDF1094, Data Platform and Sensing Technology for Environmental Sensing LAB – DPS4ESLAB.

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
