# Peer review of "A high-resolution gridded dataset of daily temperature and precipitation records (1980-2018) for Trentino – South Tyrol (northeastern Italian Alps)"

_Earth System Science Data, 2020_

## Author Comment (AC1)

This article describes the production of a high-resolution observational gridded dataset over Trentino - South Tyrol. The daily aggregated variables considered are temperature and precipitation.

The article is well structured and the presentation is clear and concise. The Introduction highlights the benefit of the study and includes a good review of the relevant literature on the topic. "Data and Methods" describes the study area and the observational database in a satisfactory manner. The interpolation scheme presented builds on a classical two-step approach. First, the climatologies are generated, then the authors use daily anomalies in their spatial analysis scheme, based on the underlying assumption of working with more Gaussian random fields. The "results and discussion" section includes the evaluation and presents a number of significant examples. As far as I can judge, there are no major flaws in the statistical analysis and the conclusions are well supported by the results. The accuracy and precision of the results are reasonable and comparable to state-of-the-art products in the Alps.

The presented method is not particularly original, because it has been applied before in the Alps, as the authors points out. The merit of this work is in the careful application of the method at such a high spatial resolution (250 m!) over complex terrain and with a pretty dense observational network. Furthermore, the final dataset is publicly available and this is a great merit of the authors. If the authors will regularly update this dataset, as they mention in their future plans (line 420), I can foresee a bright future for this dataset, which could be the basis for environmental applications and research in that part of the Alps. In conclusion, the study is valuable. My advice to the editor is to publish it (almost) as it is. I just have a few comments.

We thank the reviewer for the positive overview of our manuscript and for appreciating the method and result description. As marked by the reviewer, the aim of the work was not to test a new interpolation method but to employ an approach already proved in the Alpine context to derive a dataset which can be easily adopted for applicative purposes requiring local-scale information. For this reason, as stated in the manuscript, the dataset will be updated with a certain frequency in order to assure its availability for local studies.

We thank for the useful points raised by the reviewer. We address the comments here below and can integrate them in a revised version.

Comments:

Figure 8, total precipitation climatology. The figure shows that the elevation has an effect on the spatial distribution of precipitation. However, it looks like the distance from the sea (or from the Po plain) also has an effect on this variable. Have the authors considered to include this variate (i.e. distance from the sea) in their study?

In the method we applied in the study, the precipitation climatologies are interpolated by performing a local linear regression with elevation, which is then considered as unique predictor for precipitation distribution. The other geographical influences are included in the local selection of the stations entering the fit and in their corresponding weight. The station selection for the linear fit is in fact performed month-by-month for each grid point separately and

identifies the sites with the most similar geographical conditions to the point under reconstruction. Each station is assigned a weight which is proportional to such similarity and defines its contribution in the local linear fit. In the scheme here presented, we used distance, elevation difference and slope difference (orientation and steepness) as weighting factors. As correctly underlined by the reviewer, the sea proximity is another geographical factor which could have an influence on climate distribution.

The sea distance of the stations in the database ranges from 80 to 230 km. If we add the difference in sea proximity as further weighting factor, no improvement is observed in the reconstruction accuracy. It is worth noting that in the study area, sea distance is closely related to latitude and the similarity in geographical position is already taken into account in local station selection and contribution to the linear fit by the weight depending on the radial distance from the point under evaluation.

For these reasons we considered the sea distance factor negligible in the interpolation scheme. However, we can explicitly mention and discuss the choice in the Section 2.3.1 in a possible reviewed version of the manuscript.

It is not clear if this dataset can be used to extract climatological trends of temperature and/or precipitation. Considering the construction of the observational dataset used, I think the answer may be positive. However, given the importance of such an application, I would recommend discussing this point explicitly in the text. Perhaps, the right place where to discuss this issue would be at the end of Sec 3.1 (right after the related discussion on Fig. 7), with a further reference in the conclusions.

We thank the reviewer for highlighting the point. The reconstruction methodology and the quality check applied to the series, including the homogeneity evaluation for the longest records, make the dataset useable to analyze temperature and precipitation trends over the recent decades. Keeping the product updated is also thought to build a historical archive of climate information supporting assessment of changes and trends over the study region. However, the regional trend evaluation is beyond the scope of the submitted manuscript and the topic will be addressed in forthcoming studies employing the dataset here presented.

As suggested by the reviewer, the potential use of the dataset for trend analysis can be mentioned at the end of Section 3.1 as integration to the discussion on Figure 7. The similar magnitude of the leave-one-out errors over the whole spanned period, especially for temperature fields, demonstrates the general accuracy of the reconstructed temporal signal over the domain. We can also remark such further potential use of the dataset in the conclusions by integrating the lines 399-404 in the current version.

---

## Author Comment (AC2)

The manuscript "A high-resolution gridded dataset of daily temperature and precipitation records (1980 – 2018) for Trentino – South Tyrol (northeastern Italian Alps)" submitted by Crespi et al., presents a novel and well prepared precipitation and temperature gridded dataset for the Region Trentino-South Tyrol in North-East Italy. The main strengths of the dataset in comparison to previous works are the spatial resolution, the temporal extension up to 2018 and the aim of the authors of keeping it up to date.

I did not find particular issues in the proposed methodology and the dataset is publicly available. I consider therefor the work of interest for the readers of the journal and useful for the scientific community. Some minor comments and suggestions are listed below:

We gratefully thank the reviewer for appreciating our work on the presented dataset and remarking the interest of the manuscript for the readers. We address the minor comments and suggestions here below and they can be integrated, accordingly, in a revised version of the manuscript.

1. Some more information (e.g., resolution, accuracy, measurement error) about the quality of measured temperature and precipitation time series would be important to better appreciate the quality of the interpolated results. In fact, it seems that the interpolation error is in the same order of the measurement error, which is a nice attribute of the dataset.

The weather station measurements are affected by a number of both systematic and random errors, also depending on the type of instrumentation and the way measurements are recorded, i.e. manually or automatically. We have already mentioned in the submitted manuscript (lines 272-275) the fact that rain gauges are usually affected by systematic underestimation of precipitation, which can be larger in case of snowfall and could account for up to several tens of percent of the measured values, especially at the high-level sites characterized by higher wind speed. Due to the different data sources, different instruments and measurement protocols of the collected database, it is not possible to generalize the information on temperature and precipitation measurement errors. Nevertheless, we can improve the details on the collected data by retrieving some information from the data providers to integrate in the Section 2.2, such as for example measurement accuracy of the deployed weather stations. In addition, we can add in a revised version of the manuscript the results we obtained in the quality-check process where we simulated each measured series by using the surrounding stations and an anomaly-based scheme (lines 154-155). The comparison of simulated and measured data allowed us to highlight those series affected by the largest uncertainty but also to get a measure of the general accuracy of the records.

2. I did not get why the dataset was compared with snow-cover maps instead of (for example) other gridded products (e.g., Adler et al., 2015) or remote sensing products. P and T datasets for large parts of the region in fact were investigated in recent works such as:

Mei, Y., Anagnostou, E.N., Nikolopoulos, E.I., Borga, M., 2014. Error analysis of satellite precipitation products in mountainous basins. J. Hydrometeorol. 15, 1778–1793.

Duan, Z., Liu, J.Z., Tuo, Y., Chiogna, G., Disse, M., 2016. Evaluation of eight high spatial resolution gridded precipitation products in Adige Basin (Italy) at multiple temporal and spatial scales. Sci. Total Environ

Maybe the authors could better justify this choice and/or they may find these references useful for section 2.1.

The aim of the comparison was not to further validate the accuracy of the gridded temperature and precipitation products but to show an example of their application in combination with another parameter, for instance snow cover, derived from remotely-sensed measurements at a very fine spatial resolution. We can improve on describing our motivation for this comparison in the manuscript. The inter-comparison with other existing and coarser resolution products, such as the ones analyzed in the suggested references, would be of high interest but it requires a much more extended evaluation and discussion which likely goes beyond the scope of the current manuscript proposed as data paper. The dataset evaluation in an integrated inter-comparison analysis with a pool of rain-gauge and satellite-based products over the study area can be addressed in a future study and this can be explicitly stated in the conclusions. We therefore found the references suggested by the reviewer very useful and we can integrate them in the list of previous studies and available datasets covering the region reported in the Introduction (Section 1).

3. Line 71 please specify which local gradients you mean

We can specify "local climate gradients".

4. Lines 113-116 since the focus is on precipitation and temperature, and the area investigated is larger than the Adige basin itself, I think these lines could be removed.

We agree with the reviewer and the lines can be removed in a revised version of the manuscript.

5. The correction of 48 precipitation time series gives a particular relevance of this step to the entire process, in my view. Some more information about how the correction factors were applied from monthly to daily time series and how large were the applied adjustments would be interesting.

The adjustment of precipitation data is multiplicative and the monthly factors were directly applied to the corresponding daily values. The homogenization of temperature series was performed by additive corrections. In this case, the 12 monthly factors estimated from the monthly temperature series for the tested period were then interpolated to a daily resolution by means of a second-order trigonometric regression in order to account for the annual seasonality and to obtain an additive correction for each calendar day. As requested by the reviewer, we can improve the description of the homogenization procedure in Section 2.2 in a revised version of the manuscript. In particular, we can better explain how the monthly corrections were computed and how they were transferred to the daily series. In addition to the technical details, we can also report the number of detected breaks and the mean magnitude of the adjustments performed on precipitation and temperature series.

6. I suggest to improve figure 3 providing also information about the relative areal contribution of each elevation range. For example, a second x-axis with the cumulative area of the studied region.

We thank the reviewer for this useful suggestion. We can modify Figure 3 by adding the comparison of the relative elevation distribution of the DEM for the study region.

7. Sections 2.3.1 and 2.3.2 are a bit difficult to follow. I understand that providing too much mathematical details in the main text would make it probably too long, but in my view an appendix with a more rigorous description of the procedure would be beneficial.

Since the methodology description and the dataset evaluation are the focus of the manuscript, we would prefer not to move any text related to the method from these sections to an appendix. However, we can make the section 2.3 clearer and improve here the methodology description as requested by the reviewer.

8. Figure 5, the color-code to interpret the heat map is missing.

We can add the color legend in Figure 5.

---

## Author Response (AR1)

**Reviewer #1**

This article describes the production of a high-resolution observational gridded dataset over Trentino - South Tyrol. The daily aggregated variables considered are temperature and precipitation.

The article is well structured and the presentation is clear and concise. The Introduction highlights the benefit of the study and includes a good review of the relevant literature on the topic. "Data and Methods" describes the study area and the observational database in a satisfactory manner. The interpolation scheme presented builds on a classical two-step approach. First, the climatologies are generated, then the authors use daily anomalies in their spatial analysis scheme, based on the underlying assumption of working with more Gaussian random fields. The "results and discussion" section includes the evaluation and presents a number of significant examples. As far as I can judge, there are no major flaws in the statistical analysis and the conclusions are well supported by the results. The accuracy and precision of the results are reasonable and comparable to state-of-the-art products in the Alps.

The presented method is not particularly original, because it has been applied before in the Alps, as the authors points out. The merit of this work is in the careful application of the method at such a high spatial resolution (250 m!) over complex terrain and with a pretty dense observational network. Furthermore, the final dataset is publicly available and this is a great merit of the authors. If the authors will regularly update this dataset, as they mention in their future plans (line 420), I can foresee a bright future for this dataset, which could be the basis for environmental applications and research in that part of the Alps. In conclusion, the study is valuable. My advice to the editor is to publish it (almost) as it is. I just have a few comments.

We thank the reviewer for the positive overview of our manuscript and for appreciating the method and result description. As marked by the reviewer, the aim of the work was not to test a new interpolation method but to employ an approach already proved in the Alpine context to derive a dataset which can be easily adopted for applicative purposes requiring local-scale information. For this reason, as stated in the manuscript, the dataset will be updated with a certain frequency in order to assure its availability for local studies.

We thank for the useful points raised by the reviewer. We address the comments here below and integrated them in the revised version of the manuscript.

Comments:

Figure 8, total precipitation climatology. The figure shows that the elevation has an effect on the spatial distribution of precipitation. However, it looks like the distance from the sea (or from the Po plain) also has an effect on this variable. Have the authors considered to include this variate (i.e. distance from the sea) in their study?

In the method we applied in the study, the precipitation climatologies are interpolated by performing a local linear regression with elevation, which is considered the sole predictor for precipitation distribution. Other geographical influences are indirectly included by the local

selection of the stations entering the fit and in their corresponding weight. The station selection for the linear fit is in fact performed month-by-month for each grid point separately and identifies the sites with the most similar geographical conditions to the point under reconstruction. Each station is assigned a weight which is proportional to such similarity and defines its contribution in the local linear fit. In the scheme here presented, we used horizontal distance, elevation difference and slope difference (orientation and steepness) as weighting factors. As correctly underlined by the reviewer, the sea proximity is another geographical factor which could have an influence on climate distribution.

The sea distance of the stations in the database ranges from 80 to 230 km. When we added the difference in sea proximity as further weighting factor, no improvement was observed in the reconstruction accuracy. It is worth noting that in the study area, sea distance is closely related to latitude and the similarity in geographical position is already taken into account in local station selection and contribution to the linear fit by the weight depending on the radial distance from the point under evaluation.

For these reasons we considered the sea distance factor negligible in the interpolation scheme. However, we explicitly mentioned and discussed this choice in Section 2.3.1 in the revised version of the manuscript at lines 250-252 (version without track-changes). We also improved the overall method description as suggested by the other reviewer.

It is not clear if this dataset can be used to extract climatological trends of temperature and/or precipitation. Considering the construction of the observational dataset used, I think the answer may be positive. However, given the importance of such an application, I would recommend discussing this point explicitly in the text. Perhaps, the right place where to discuss this issue would be at the end of Sec 3.1 (right after the related discussion on Fig. 7), with a further reference in the conclusions.

We thank the reviewer for highlighting the point. The reconstruction methodology and the quality check applied to the series, including the homogeneity evaluation for the longest records, make the dataset useful for the analysis of temperature and precipitation trends over the recent decades. Keeping the product updated is also thought to build a historical archive of climate information supporting the assessment of the spatio-temporal variability over the study region. However, the regional trend evaluation is beyond the scope of the submitted manuscript and the topic will be addressed in forthcoming studies.

As suggested by the reviewer, this potential use of the dataset for trend analysis is now mentioned at the end of Section 3.1 at lines 342-346 right after the discussion on Figure 7. The similar magnitude of the leave-one-out errors over the whole spanned period, especially for temperature fields, supports the general accuracy of the reconstructed temporal signal over the domain. However, we also remarked the decrease of data coverage at high-elevation areas where outcomes need to be treated cautiously. We also recalled the availability of the dataset for trend analyses at lines 458-462 in the Conclusions of the revised manuscript.

**Reviewer #2**

The manuscript "A high-resolution gridded dataset of daily temperature and precipitation records (1980 – 2018) for Trentino – South Tyrol (northeastern Italian Alps)" submitted by Crespi et al., presents a novel and well prepared precipitation and temperature gridded dataset for the Region Trentino-South Tyrol in North-East Italy. The main strengths of the dataset in comparison to previous works are the spatial resolution, the temporal extension up to 2018 and the aim of the authors of keeping it up to date.

I did not find particular issues in the proposed methodology and the dataset is publicly available. I consider therefor the work of interest for the readers of the journal and useful for the scientific community. Some minor comments and suggestions are listed below:

We gratefully thank the reviewer for appreciating our work on the presented dataset and remarking the interest of the manuscript for the readers. We provided our responses to the minor comments and suggestions here below by reporting the related changes we made in the revised version of the manuscript.

1. Some more information (e.g., resolution, accuracy, measurement error) about the quality of measured temperature and precipitation time series would be important to better appreciate the quality of the interpolated results. In fact, it seems that the interpolation error is in the same order of the measurement error, which is a nice attribute of the dataset.

The weather station measurements are affected by a number of both systematic and random errors, also depending on the type of instrumentation and the way measurements are recorded, i.e. manually or automatically. Due to the different data sources, the different instruments and measurement protocols relative to the collected database, it is not straightforward to generalize the information on temperature and precipitation measurement errors.

However, at the beginning of the Section 2.2 at lines 135-138 (version without track-changes) of the revised manuscript we specified that the measurements can be affected by a number of the different sources of errors which make crucial the quality and homogeneity controls. This aspect was also recalled at lines 320-327 where we specifically discussed the underestimation of precipitation by rain gauges, which can be larger in case of snowfall and could account for up to several tens of percent of the measured values, especially at the high-level sites characterized by higher wind speed.

In order to support the input data assessment with more quantitative information on the overall quality of the station observations, we extended the description of the spatial consistency analysis at lines 161-168. In this procedure, all $T_{max}$, $T_{min}$ and P series were simulated by using the surrounding stations and the comparison with the observations provided further insights on the station data accuracy. In particular, we reported in the manuscript for each variable the resulting bias, MAE and squared correlation as averages over all tested series.

2. I did not get why the dataset was compared with snow-cover maps instead of (for example) other gridded products (e.g., Adler et al., 2015) or remote sensing products. P and T datasets for large parts of the region in fact were investigated in recent works such as:

Mei, Y., Anagnostou, E.N., Nikolopoulos, E.I., Borga, M., 2014. Error analysis of satellite precipitation products in mountainous basins. J. Hydrometeorol. 15, 1778–1793.

Duan, Z., Liu, J.Z., Tuo, Y., Chiogna, G., Disse, M., 2016. Evaluation of eight high spatial resolution gridded precipitation products in Adige Basin (Italy) at multiple temporal and spatial scales. Sci. Total Environ

Maybe the authors could better justify this choice and/or they may find these references useful for section 2.1.

The aim of the comparison was not to further validate the accuracy of the gridded temperature and/or precipitation products but to show an example of their application in combination with another parameter, for instance snow cover, derived from remotely-sensed measurements at a similar fine spatial resolution. We restated the sentences at lines 103-105 in the Introduction, at lines 407-409 in Section 3.2  and at lines 478-479 in the Conclusions in order to remark that the aim of the comparison is to provide an example of combination of the dataset with another parameter from a fine spatial resolution data source.

The inter-comparison with other existing and coarser resolution products, such as the ones analyzed in the suggested references, would be of high interest but it requires a much more extended evaluation and discussion which likely goes beyond the scope of the current manuscript proposed as data paper. The dataset evaluation in an integrated inter-comparison analysis with a pool of rain-gauge and satellite-based products over the study area can be addressed in a future study and this was explicitly stated in the conclusions at lines 471-474.

We therefore found the references suggested by the reviewer very useful and we decided to mention in the Introduction this dataset category as well as the previous evaluation analyses performed over large portions of the study region (lines 49-55).

3. Line 71 please specify which local gradients you mean

We specified "local climate gradients" at line 77.

4. Lines 113-116 since the focus is on precipitation and temperature, and the area investigated is larger than the Adige basin itself, I think these lines could be removed.

We removed the lines, however we decided to keep the description of the main valley distribution at lines 118-120 since it is a relevant feature for interpreting the climate of the region.

5. The correction of 48 precipitation time series gives a particular relevance of this step to the entire process, in my view. Some more information about how the correction factors were applied from monthly to daily time series and how large were the applied adjustments would be interesting.

In case of precipitation, the adjustment was multiplicative and the monthly factors were directly applied to the corresponding daily values. The homogenization of temperature series was conversely performed by additive corrections. In this case, the 12 monthly correction factors estimated from the monthly temperature data for the tested period were interpolated to the daily resolution by means of a second-order trigonometric interpolation in order to account for the annual seasonality and to obtain an additive correction for each calendar day. By following the suggestion, we improved the description of the homogenization procedure in Section 2.2 at lines 170-184 and we added some references in which a similar method was discussed and applied. In particular, we specified how the monthly corrections were estimated and how they were transferred to daily series for both temperature and precipitation.

In addition to the method description, we included a summary of the applied corrections at lines 185-191. In particular, we reported for each variable the number of breakpoints, the average length of corrected periods and the annual average of the corrections. We also specified that a relevant portion of adjusted years was prior to 1980, which is before the period of the generated spatial product.

6. I suggest to improve figure 3 providing also information about the relative areal contribution of each elevation range. For example, a second x-axis with the cumulative area of the studied region.

We thank the reviewer for this useful suggestion. We modified Figure 3 by reporting on the same plot the absolute elevation distribution (200-m bins) of both temperature and precipitation stations and by adding a secondary y-axis for the relative elevation distribution of the DEM over the study region. It is worth noting that the areas above 1500 m are undersampled with respect to the lower elevations remarking that a larger uncertainty needs to be associated to the resulting estimates for the high-elevation points. We also discussed this aspect at lines 205-211.

7. Sections 2.3.1 and 2.3.2 are a bit difficult to follow. I understand that providing too much mathematical details in the main text would make it probably too long, but in my view an appendix with a more rigorous description of the procedure would be beneficial.

Since the methodology description and the dataset evaluation are the focus of the manuscript, we preferred not to move any text related to the method from the main body to an appendix. However, we tried to improve the methodology description in Section 2.3 as suggested by the reviewer. In particular:

- we integrated the Section 2.3.1 (especially lines 245-257) by providing further details on the station weight definition and on their selection for the weighted linear fit applied in the climatology interpolation. The mathematical expression of the total station weight was also added in Eq. (2) to help the interpretation of the overall framework.

- we restated the Section 2.3.2 by adding more details on the anomaly calculation, on the station weight definition, and on the final derivation of the absolute fields. In order to make the explanation more rigorous, we added and discussed the mathematical expressions throughout the text (Eq. 4-6).

8. Figure 5, the color-code to interpret the heat map is missing.

We modified the figure by adopting a more readable color palette and by adding the legend.

**Further changes**

We included some minor changes in the revised text, which are all highlighted in the track-changes file.

We added a few additional references and updated the previous entries which have been recently published and assigned to an issue.

We updated the reference for the dataset repository by reporting the registered doi in PANGAEA.